# Influence of Different Types of β-Blockers on Mortality in Patients on Hemodialysis

**DOI:** 10.3390/biomedicines11102838

**Published:** 2023-10-19

**Authors:** Seok-Hui Kang, Bo-Yeon Kim, Eun-Jung Son, Gui-Ok Kim, Jun-Young Do

**Affiliations:** 1Division of Nephrology, Department of Internal Medicine, College of Medicine, Yeungnam University, Daegu 42415, Republic of Korea; kangkang@ynu.ac.kr; 2Healthcare Review and Assessment Committee, Health Insurance Review and Assessment Service, Wonju 26465, Republic of Korea; 3Quality Assessment Department, Health Insurance Review and Assessment Service, Wonju 26465, Republic of Korea

**Keywords:** hemodialysis, β-blocker, mortality

## Abstract

Previous results regarding the association between types of β-blockers and outcomes in patients on hemodialysis (HD) were inconsistent. Our study aimed to evaluate patient survival according to the type of β-blockers administered using a large sample of patients with maintenance HD. Our study included patients on maintenance HD patients from a national HD quality assessment program (*n* = 54,132). We divided included patients into four groups based on their use and type; Group 1 included patients without a prescription of β-blockers, Group 2 included patients with a prescription of dialyzable and cardioselective β-blockers, Group 3 included patients with a prescription of non-dialyzable and non-cardioselective β-blockers, and Group 4 included patients with prescription of non-dialyzable and cardioselective β-blockers. The number of patients in Groups 1, 2, 3, and 4 were 34,514, 2789, 15,808, and 1021, respectively. The 5-year survival rates in Groups 1, 2, 3, and 4 were 69.3%, 66.0%, 68.8%, and 69.2%, respectively. Univariate Cox regression analyses showed the hazard ratios to be 1.10 (95% CI, 1.04–1.17) in Group 2 and 1.05 (95% CI, 1.02–1.09) in Group 3 compared to Group 1. However, multivariate Cox regression analyses did not show statistical significance among the four groups. Our study showed that there was no significant difference in patient survival based on the use or types of β-blockers.

## 1. Introduction

The end-stage renal disease requires renal replacement therapy. Among the three known renal replacement therapies, namely hemodialysis (HD), peritoneal dialysis, and kidney transplantation, HD is the most commonly used modality [1,2]. Patients on HD are, however, known to have poorer clinical outcomes than those not on it [3]. The most common cause of death in patients with HD is cardiovascular disease. Many studies have focused on the prevention of or treatment options for cardiovascular diseases to prolong the survival of patients on HD; however, the incidence of cardiovascular disease in patients on HD remains high [1,2,4,5]. Patients on HD are prescribed various medications related to cardiovascular diseases. However, the effect of medications in patients with HD can be largely influenced through dialysis sessions or the medication’s characteristics compared to that in patients without dialysis. Furthermore, pharmacokinetics and/or pharmacodynamics in uremic conditions are different from those in patients not on dialysis. Therefore, even among medications within the same class, prescriptions may differ depending on the characteristics of the drug.

β-blockers are the most commonly used antihypertensive drugs in patients with HD. They are known to have survival benefits in patients with myocardial infarction (MI) or heart failure beyond simple blood pressure control [6]. However, they have different characteristics based on dialyzability and/or cardioselectivity, which may result in different effects on the survival of patients with HD. Previous studies have evaluated the association between types of β-blockers and outcomes in patients with HD [7,8,9,10,11,12,13,14,15,16,17,18]. However, results regarding this association were inconsistent. Moreover, most studies were performed using registry data from limited regions or data sources. Therefore, additional studies are needed to confirm if the type of β-blockers influences survival in patients with HD. Our study aimed to evaluate patient survival according to the type of β-blockers administered using a large sample of patients with maintenance HD.

## 2. Materials and Methods

### 2.1. Data and Study Participants

This retrospective study utilized data from a national HD quality assessment program, as well as information from claims data provided by the Health Insurance Review and Assessment (HIRA) of the Republic of Korea [19,20]. The HD quality assessment programs were conducted in three phases: the 4th (July 2013–December 2013), the 5th (July 2015–December 2015), and the 6th (March 2018–August 2018). Participants eligible for the study were patients on maintenance HD for at least three months, receiving HD at least twice a week (at least eight times per month), and aged 18 years or older. We analyzed the HD quality assessment and claims data of all patients on HD assessed by HIRA.

The total number of participants in the 4th, 5th, and 6th HD quality assessment programs were 21,846, 35,538, and 31,294, respectively (Figure 1).

From this pool, we excluded individuals who had participated in multiple assessments or had incomplete datasets, those who underwent HD via non-cuffed or cuffed tunneled catheter, and those who were prescribed two or more β-blockers or received them for less than 90 days within each six-month assessment period. Finally, our study included a total of 54,132 patients. The study received approval from the institutional review board (IRB) of the Yeungnam University Medical Center (approval no. YUMC 2022-01-010). Informed consent was not obtained from the patients as all records and participant information had been anonymized and de-identified before the analysis. 

### 2.2. Variables

We gathered a comprehensive set of data variables, including demographic information such as age, sex, primary cause of end-stage renal disease, duration of HD treatment (in months), and types of vascular access used. Our dataset also incorporated laboratory measurements taken during the assessment, such as hemoglobin levels (measured in g/dL), Kt/V_urea_, serum albumin concentration (measured in g/dL), serum calcium levels (measured in mg/dL), serum phosphorus levels (measured in mg/dL), serum creatinine levels (measured in mg/dL), predialysis systolic blood pressure readings (measured in mmHg), predialysis diastolic blood pressure readings (measured in mmHg), and ultrafiltration volume per session (measured in liters). Data collection occurred monthly, with all laboratory values averaged across the readings. Kt/V_urea_ was calculated using Daugirdas’ equation [21].

The codes for medication are shown in Appendix A. The use of β-blockers was defined as a prescription for ≥90 days during 6 months of each HD quality assessment period. β-blockers were classified into three groups according to their type, as described in previous studies [16,17,18,19,20,21]. The available oral β-blockers in the Republic of Korea are atenolol, metoprolol, bisoprolol, propranolol, carvedilol, betaxolol, and nebivolol. We divided included patients into four groups based on their use and type; Group 1 included patients without prescription of β-blockers, Group 2 included patients with prescription of atenolol, metoprolol, or bisoprolol (dialyzable and cardioselective), Group 3 included patients with prescription of propranolol or carvedilol (non-dialyzable and non-cardioselective), and Group 4 included patients with prescription of nebivolol or betaxolol (non-dialyzable and cardioselective). In our study, we classified a patient as having used aspirin, renin-angiotensin system blockers, clopidogrel, ticlopidine, or statin if they had filled and utilized one or more prescriptions for the medication within the year prior to the evaluation of the HD quality assessment program. For patients who underwent HD for less than one year, we adjusted the criterion to align with the duration of their HD treatment. Specifically, we examined whether they had filled and utilized one or more prescriptions for the medication during their HD period before the HD quality assessment program, which could be shorter than a year.

The presence of comorbidities was evaluated for a year before the evaluation of the HD quality assessment program. Comorbidity was defined using the codes utilized by Quan et al. [22,23]. The Charlson Comorbidity Index (CCI) includes 17 comorbidities. All patients in our study underwent HD and were considered to indicate the presence of renal disease. Other comorbidities and their ICD-10 codes are shown in Appendix A. Finally, the CCI score was calculated. Furthermore, we defined atrial fibrillation using I48 of the ICD-10 code.

We assessed all-cause mortality as the primary outcome and cardiac-cerebrovascular events (CVE) as the secondary outcome in our study. CVE was defined as interventions related to revascularization, which were evaluated using medical treatment, procedure, or operation codes listed in Appendix A. Mortality was evaluated for all patients. For the analysis of CVE, we excluded patients who underwent interventions associated with CVE in the year before and during HD quality assessment. 

We tracked the outcomes continuously until April 2022. In cases where a patient transferred to peritoneal dialysis or underwent kidney transplantation, the date of transfer or transplantation was marked as the endpoint of follow-up, and the data were censored accordingly. Throughout the follow-up period, electronic records were used to define clinical outcomes, excluding cases of death. Censoring codes applied were O7072, O7071, or O7061 for peritoneal dialysis and R3280 for kidney transplantation. Information regarding patient mortality was extracted from the HIRA database. 

### 2.3. Statistical Analyses

The data were subjected to analysis using the SAS Enterprise Guide version 7.1 (SAS Institute, Cary, NC, USA) or R version 3.5.1 (R Foundation for Statistical Computing, Vienna). Categorical variables were expressed in terms of numbers and percentages, while continuous variables were presented as means with accompanying standard deviations. To analyze categorical variables, Pearson’s χ^2^ test or Fisher’s exact test was employed. Continuous variable comparisons were conducted using a one-way analysis of variance, followed by the Tukey post hoc test. 

Survival estimates were computed using the Kaplan–Meier curve, and Cox regression analyses were applied. The log-rank test was employed to determine *p*-values for the comparison of survival curves. In our multivariate Cox regression analyses, adjustments were made for various factors, including age, gender, type of vascular access, the underlying cause of end-stage renal disease, CCI score, HD vintage, ultrafiltration volume, Kt/V_urea_, hemoglobin levels, serum albumin levels, serum creatinine levels, serum phosphorus levels, serum calcium levels, predialysis systolic blood pressure, predialysis diastolic blood pressure, and the use of medications such as aspirin, renin–angiotensin system blockers, clopidogrel, ticlopidine, or statins, and the presence of MI or congestive heart failure (CHF). These multivariate Cox regression analyses were conducted using the enter mode. Statistical significance was set at *p* < 0.05.

## 3. Results

### 3.1. Clinical Characteristics

The numbers of patients in Groups 1, 2, 3, and 4 were 34,514, 2789, 15,808, and 1021, respectively (Table 1). 

Male predominance was lower in Group 1 than in the other three groups. The patients in Group 1 had lower incidences of diabetes mellitus (DM), use of renin–angiotensin system blockers, statin, aspirin, clopidogrel, or ticlopidine, and MI or CHF than those in the other three groups. They had lower CCI scores, SBP, and ultrafiltration volume than those in the other groups. Moreover, the patients in Group 1 had greater Kt/V_urea_ and lower phosphorus and creatinine levels than those in Group 2 or 3. Among the four groups, arteriovenous fistulas were commonest in Group 3. Hemoglobin levels in Group 1 or 2 were greater than those in Group 3. 

### 3.2. Survival Analyses

The respective numbers of patients in the survivor, death, peritoneal dialysis, or kidney transplantation subgroup at the endpoint of follow-up were 18,407 (53.3%), 13,380 (38.8%), 119 (0.3%), and 2608 (7.6%) in Group 1; 1449 (52.0%), 1120 (40.2%), 15 (0.5%), and 205 (7.4%) in Group 2; 8285 (52.4%), 6249 (39.5%), 56 (0.4%), and 1218 (7.7%) in Group3; and 565 (55.3%), 388 (38.0%), 0, and 68 (6.7%) in Group 4 (*p* = 0.113).

The 5-year survival rates in Groups 1, 2, 3, and 4 were 69.3%, 66.0%, 68.8%, and 69.2%, respectively (Figure 2). 

Group 1 exhibited superior patient survival rates compared to both Group 2 and 3. Univariate Cox regression analyses revealed hazard ratios of 1.10 (95% CI, 1.04–1.17) for Group 2 and 1.05 (95% CI, 1.02–1.09) for Group 3 compared to Group 1 (Table 2). 

However, multivariate Cox regression analyses did not show statistical significance among the four groups. We also performed Cox regression analyses based on the characteristics of β-blockers (Table 3). 

Univariate and multivariate Cox regression analyses revealed consistent patterns, as observed in the analysis of the four distinct groups.

Subsequently, we conducted subgroup analyses stratified by sex, age, and the presence of DM and heart disease (MI or CHF) (Appendix A). In these subgroup analyses, most multivariate analyses did not show a significant association between groups and mortality, similar to the findings of the overall study cohort. Nevertheless, among patients of advanced age, those with DM, or individuals with heart disease, Group 3 exhibited superior patient survival compared to Group 2. Furthermore, we performed subgroup analyses using patients with MI, CHF, or atrial fibrillation. The majority of results using these subgroups indicated that the use or types of β-blockers were not associated with mortality.

### 3.3. CVE Analyses

For CVE analyses of CVE, there were 31,080 patients in Group 1, 2353 patients in Group 2, 14,129 patients in Group 3, and 887 patients in Group 4. Appendix A shows the Kaplan–Meier curves of CVE according to the groups.

The 5-year survival rates in Groups 1, 2, 3, and 4 were 86.3%, 83.6%, 85.1%, and 85.1%, respectively. On univariate and multivariate analyses, Group 2 had poorer CVE-free survival rates than Group 1 or 3 (Appendix A). Group 2 showed a trend of poorer CVE-free survival rates than did Group 4, although it was not statistically significant.

## 4. Discussion

We analyzed 54,132 patients who underwent an HD quality assessment program in the Republic of Korea. Our results revealed that the use of β-blockers was not associated with mortality in patients on maintenance HD. In univariate Cox regression analyses, Group 2 or 3 had poorer patient survival than Group 1. However, in multivariate Cox regression analyses, statistical significance was not seen across all four groups. Subgroup analyses using patients with old age, DM, or heart diseases showed a favorable patient survival in Group 3 than in Group 2. 

Appendix A shows the summary of previous studies regarding the association between types of β-blockers and outcomes in patients with HD; inconsistent results were found to exist. Wu et al. analyzed a registry from Taiwan and showed that bisoprolol as a dialyzable and cardioselective β-blocker had better patient outcomes than carvedilol as a non-dialyzable and non-cardioselective β-blocker [7]. Two studies using the United States Renal Data System showed that atenolol or metoprolol as dialyzable and cardioselective β-blockers had better patient survival and cardiovascular events than carvedilol or labetalol as non-dialyzable and non-cardioselective β-blockers [8,9]. However, another study using a registry from Taiwan showed similar patient survival across all three types of β-blockers [10]. Wier et al. revealed that non-dialyzable β-blockers are associated with better clinical outcomes than dialyzable ones [12]. 

Some hypotheses are associated with favorable outcomes regarding cardioselective β-blockers compared to non-cardioselective ones. Cardioselective β-blockers lower blood pressure via a decrease in cardiac output rather than vasodilation; however, drugs with affinity to α- and β-receptors can attenuate compensatory vasoconstriction, which is associated with hemodynamic instability during ultrafiltration [8,9]. β1-selective blockers lead to an increase in β1-receptor sensitivity and density, which further results in a better response to adrenergic stimuli [22]. In addition, drugs with affinity to β2 receptors can decrease potassium influx within cells and influence the development of hyperkalemia [23]. Furthermore, the removal of medications during dialysis can attenuate intradialytic hypotension and may be associated with favorable outcomes in patients with dialyzable β-blockers. 

Some studies showed better clinical outcomes with dialyzable and/or cardioselective β-blockers than with non-dialyzable and/or non-cardioselective ones, despite overall inconsistent results between the types of β-blockers and the clinical outcomes [7,8,9,10,11,12,13,14,15,16,17,18]. However, our study did not show superiority in patient survival based on the type of β-blockers. Intake of the medication after an HD session or cessation of medication on the day of HD may attenuate the beneficial effect of dialyzable and/or cardioselective β-blockers on intradialytic hypotension. Additionally, intradialytic hypotension can easily develop in patients with large weight gains, although this could simply be an indicator of nutritional status. Good nutritional status in patients with large weight gains may offset the hazardous effects of intradialytic hypotension. Furthermore, patients on cardioselective β-blockers may develop heart block or bradycardia more easily due to hyperkalemia or other electrolyte abnormalities. 

In subgroups with old age, DM, or heart diseases, non-dialyzable and non-cardioselective β-blockers were associated with better patient survival than dialyzable and cardioselective ones. The risk of hyperkalemia is greater in such patients, and they are more susceptible to changes in pharmacokinetics during HD sessions. Therefore, non-dialyzable and non-cardioselective β-blockers would be helpful in maintaining stable blood pressure via stable pharmacokinetics during dialysis in stable patients on HD; the low risk of bradycardia with non-cardioselective β-blockers may be useful in attenuating heart blocks or bradycardias due to electrolyte abnormalities.

The prognosis of patients with HD can be significantly influenced by various comorbidities; therefore, adjustment for these factors is essential in multivariate analyses. In our study, instead of adjusting individually for each comorbidity, we chose a simplified approach. Specifically, we utilized a merged score from the Charlson Comorbidity Index (CCI) as a confounding factor in multivariate analyses and independently analyzed DM and heart diseases (MI or CHF). The CCI in our study included 17 comorbidities: MI, CHF, peripheral vascular disease, cerebrovascular disease, dementia, chronic pulmonary disease, rheumatologic disease, peptic ulcer disease, mild liver disease, DM without chronic complications, hemiplegia, renal disease, DM with chronic complications, any malignancy, moderate to severe liver disease, metastatic tumor, and acquired immune deficiency syndrome. Among these comorbidities, only heart disease (MI or CHF) was considered an independent confounding factor or utilized in subgroup analysis. For DM, it was only used in the subgroup analysis. However, for other comorbidities, we chose to include them solely as confounding factors in multivariate analyses, utilizing the CCI scores derived from them. While it is possible that each comorbidity included in the CCI could individually affect the prognosis of patients with HD, our study aimed to primarily examine differences in overall mortality rates based on types of β-blockers. Therefore, we selected a simpler model. Analyzing the individual risks associated with each comorbidity is an interesting research topic but beyond the scope of our study’s main objectives.

In the analyses of patients not receiving treatment for recent cardiac or cerebrovascular events, the CVE-free survival rate was poorer in Group 2 than in Group 1 or 3, even though there was no association with all-cause mortality. Further, Group 2 had a poorer CVE-free survival trend than did Group 4. The lack of statistical significance between Group 2 and Group 4 may be due to the small sample size in Group 4. These findings are consistent with Weir’s study, which demonstrated favorable outcomes with non-dialyzable β-blockers compared to dialyzable β-blockers [12]. These suggest that, for patients without recent cardiac or cerebrovascular interventions, the benefits of dialyzable β-blockers for intradialytic hypotension were relatively attenuated, and hazard effects, such as an increase in hypertension after removal of the drug during HD, may be augmented. Patients without recent cardiac or cerebrovascular interventions can be prescribed β-blockers due to simple blood pressure. In addition, non-cardioselective β-blockers have a greater effect on lowering blood pressure than cardioselective β-blockers. Removal of the drug during HD could worsen blood pressure during the HD session and potentially increase the risk of CVE, especially for those taking dialyzable β-blockers. However, it is important to consider the limitations of our study and the lack of significant association with all-cause mortality when interpreting these results.

We conducted subgroup analyses to investigate the effects of β-blockers and potential differences among β-blocker types in patients with coexisting conditions, such as MI, CHF, and atrial fibrillation. However, our study did not demonstrate a mortality benefit associated with β-blocker use or a significant difference among β-blocker types. Based on previous research and considering the specific limitations and characteristics of our study, we expected that β-blocker use in patients on HD with conditions like MI, CHF, and atrial fibrillation would likely benefit patient prognosis. The lack of such findings in our study may be attributed to certain limitations and patient characteristics. First, we acknowledge that using ICD-10 codes for diagnostic accuracy might have introduced inaccuracies since accurate diagnosis usually involves essential tests such as coronary angiography, electrocardiography, and echocardiography. Moreover, the high incidence of comorbidities and the relatively advanced age of patients on HD significantly affected the potential side effects of β-blockers, potentially diminishing their overall benefits. Hence, when contemplating the use of β-blockers in patients with HD with suitable indications, it is crucial to thoroughly evaluate and, if necessary, adjust their usage considering associated complications.

Summarizing our study’s findings in the context of β-blocker use, type differences, analyses across various subgroups, and existing literature, we conclude that patients on HD exhibit unique variations. However, there is currently insufficient evidence to conclusively demonstrate the superiority of a specific β-blocker type. Therefore, decisions regarding β-blocker use should be based on pharmacodynamics and a personalized approach. While β-blockers, particularly cardioselective ones, are beneficial in conditions such as MI, CHF, and atrial fibrillation, it is important to consider factors like blood pressure fluctuations, heart rate, and the timing of these changes when deciding whether to continue using or switch β-blocker types. Furthermore, if the primary goal is blood pressure control, non-cardioselective β-blockers could be prioritized. In patients on HD, where the benefits may not be significantly greater than those in non-dialysis patients due to various underlying conditions, it is reasonable to not be overly cautious about discontinuing β-blockers if they cause side effects that could potentially worsen prognosis. However, we did observe differences in mortality rates related to the type of β-blockers in specific subgroups, especially in older adults, those with DM, and heart diseases. In these groups, non-dialyzable non-cardioselective β-blockers were associated with a more favorable prognosis compared to dialyzable β-blockers. This may be due to lower hemodynamic changes.

Our study had some limitations. First, the study was designed retrospectively, and there existed a substantial disparity in sample size and baseline characteristics across the four groups. The retrospective nature of the study may lead to selection bias, data quality concerns, and challenges in establishing causal relationships. Therefore, caution should be exercised when generalizing findings or interpreting the results. Second, comorbidities, including heart diseases and the use of β-blockers, were evaluated using claims data. Our study lacks data on precise medication dosages, timing of administration related to dialysis sessions, and medication self-adjustment based on blood pressure changes. There may be discrepancies between the prescribed medication amount and the actual dosage taken, particularly among dialysis patients who may take medications after dialysis due to intradialytic hypotension or immediately before dialysis in cases of intradialytic hypertension. Patients may voluntarily discontinue medications when blood pressure is low or take additional doses when it is high. We should consider these factors as significant confounding variables that could potentially affect our results, but the lack of detailed information on medication administration practices is a limitation of our study. In addition, the etiology of use of β-blockers drug intake can influence patient outcomes. Effects of β-blockers on patient survival may be different between patients taking medications for MI or CHF and those taking medications for simple blood pressure control. Furthermore, the timing of drug intake, such as before or after an HD session or cessation of medication on the day of HD, can influence the association between β-blockers types and patient outcomes. Third, our study lacks data on the cause of death, the occurrence of intradialytic hypotension, and supplementary measurements related to heart function, such as ejection fraction, left ventricular hypertrophy, heart rate, or cardiac mass. β-blockers are known to be effective in managing cardiovascular diseases, and having information on cardiovascular-related mortality or heart function would be highly valuable in understanding the specific advantages and potential risks associated with different types of β-blockers types. This information goes beyond just assessing overall mortality. The presence of intradialytic hypotension is especially considered one of the main hazardous effects of non-dialyzable and/or non-cardioselective β-blockers and data in that regard would be useful to understand the cause-relationship between β-blocker types and patient death. Additionally, another important limitation of our study is that we did not consider the cause and severity of heart failure. We lacked data on these aspects, which is inherent to our study design. Our study diagnoses the disease using a different code-based approach instead of relying on chart review, laboratory, and imaging data. Our approach to heart failure in this study was limited in terms of detailed cause assessment and the ability to assess the degree of functional impairment. Heart failure has various potential causes, such as ischemia, blood pressure, arrhythmia, or amyloidosis, and can be categorized into three groups of heart failure with reduced, moderately reduced, and preserved rejection fractions. These factors can impact prognosis and clinical outcomes differently depending on the type of β-blockers used. We acknowledge that this limitation is significant in our study. We believe that addressing these limitations will be crucial in future research by conducting comprehensive subgroup analyses and utilizing other methods.

## 5. Conclusions

Our study showed that there was no significant difference in patient survival based on the use or types of β-blockers. Subgroups with old age, DM, or heart diseases had survival benefits with non-dialyzable and non-cardioselective β-blockers than with dialyzable and cardioselective ones. In addition, patients who did not receive interventions associated with revascularization had a better CVE-free survival trend with non-dialyzable than with dialyzable ones. Our results should be carefully interpreted, considering the aforementioned limitations of this study. Additional randomized prospective investigations are necessary to establish definitive conclusions regarding the impact of β-blocker types on outcomes among patients undergoing HD.

## Figures and Tables

**Figure 1 biomedicines-11-02838-f001:**
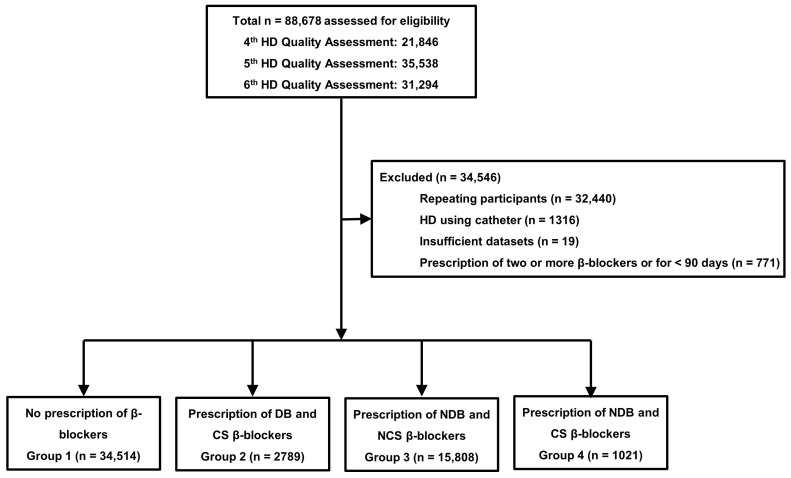
Study flow chart. Abbreviations: CS, cardioselective; DB, dialyzable; HD, hemodialysis; NCS, non-cardioselective; NDB, non-dialyzable.

**Figure 2 biomedicines-11-02838-f002:**
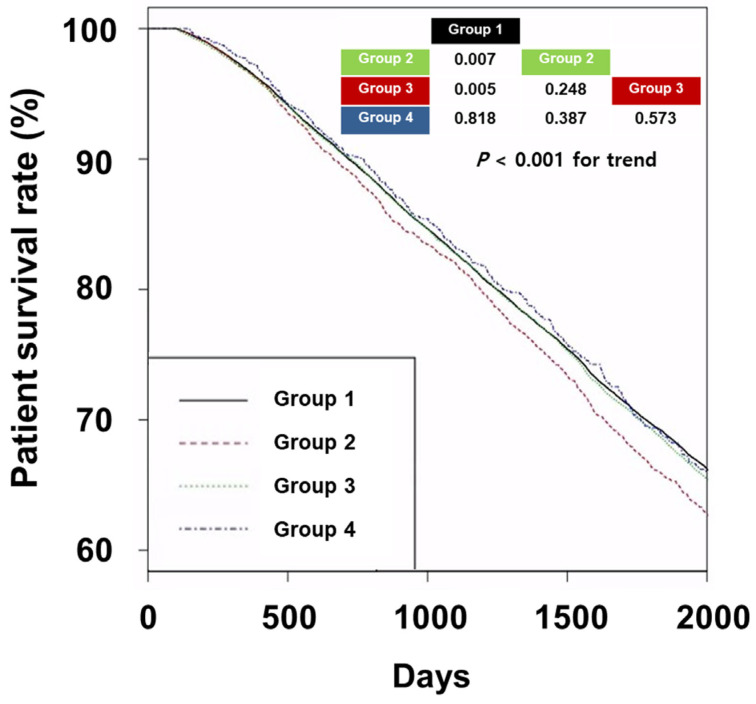
Kaplan–Meier curves of patient survival according to groups. The *p*-values for pairwise comparison or trend with log-rank tests are added to the upper right corner of the graph. Abbreviations: Group 1, patients without prescription of β-blockers; Group 2, patients with prescription of dialyzable and cardioselective β-blockers; Group 3, patients with prescription of non-dialyzable and non-cardioselective β-blockers; Group 4, patients with prescription of non-dialyzable and cardioselective β-blockers.

**Table 1 biomedicines-11-02838-t001:** Patient clinical characteristics.

	Group 1 (*n* = 34,514)	Group 2 (*n* = 2789)	Group 3 (*n* = 15,808)	Group 4 (*n* = 1021)	*p*
Age (years)	60.5 ± 13.3	60.5 ± 12.5	59.7 ± 12.4 *^#^	60.3 ± 12.3	<0.001
Sex (male, %)	19,753 (57.2%)	1737 (62.3%)	10,157 (64.3%)	647 (63.4%)	<0.001
Hemodialysis vintage (months)	53 ± 58	51 ± 56	50 ± 51 *	44 ± 49 *^#+^	<0.001
Underlying causes of ESRD					<0.001
Diabetes mellitus	13,942 (40.4%)	1365 (48.9%)	7957 (50.3%)	504 (49.4%)	
Hypertension	9095 (26.4%)	681 (24.4%)	4152 (26.3%)	277 (27.1%)	
Glomerulonephritis	3925 (11.4%)	266 (9.5%)	1438 (9.1%)	96 (9.4%)	
Others	3319 (9.6%)	199 (7.1%)	958 (6.1%)	67 (6.6%)	
Unknown	4233 (12.3%)	278 (10.0%)	1303 (8.2%)	77 (7.5%)	
CCI score	7.3 ± 2.9	7.9 ± 2.9*	7.8 ± 2.8 *	7.9 ± 2.6 *	<0.001
Follow-up duration (months)	62 ± 29	59 ± 28*	60 ± 28 *	60 ± 27	<0.001
Type of vascular access					<0.001
Arteriovenous fistula	29,249 (84.7%)	2366 (84.8%)	13,658 (86.4%)	866 (84.8%)	
Arteriovenous graft	5265 (15.3%)	423 (15.2%)	2150 (13.6%)	155 (15.2%)	
Kt/V_urea_	1.54 ± 0.27	1.51 ± 0.27^*^	1.51 ± 0.27 *	1.52 ± 0.30	<0.001
Ultrafiltration volume (L/session)	2.22 ± 0.97	2.28 ± 0.94 *	2.38 ± 0.92 *^#^	2.33 ± 0.92 *	<0.001
Hemoglobin (g/dL)	10.7 ± 0.8	10.7 ± 0.7	10.6 ± 0.7 *^#^	10.6 ± 0.8 *	<0.001
Serum albumin (g/dL)	3.99 ± 0.34	3.98 ± 0.34	3.99 ± 0.34 ^#^	3.98 ± 0.33	0.027
Serum phosphorus (mg/dL)	4.94 ± 1.37	5.05 ± 1.43 *	5.01 ± 1.35 *	4.93 ± 1.36	<0.001
Serum calcium (mg/dL)	8.90 ± 0.84	8.89 ± 0.81	8.90 ± 0.80	8.77 ± 0.75 *^#+^	<0.001
Systolic blood pressure (mmHg)	140 ± 16	143 ± 15 *	144 ± 14 *^#^	142 ± 14 *^+^	<0.001
Diastolic blood pressure (mmHg)	78 ± 9	77 ± 10 *	79 ± 9 *^#^	77 ± 10 *^+^	<0.001
Serum creatinine (mg/dL)	9.4 ± 2.8	9.5 ± 2.7 *	9.7 ± 2.6 *^#^	9.5 ± 2.8	<0.001
Use of RASB	7932 (23.0%)	1022 (36.6%)	7016 (44.4%)	443 (43.4%)	<0.001
Use of statin	8976 (26.0%)	1103 (39.5%)	5487 (34.7%)	415 (40.6%)	<0.001
Use of aspirin	13246 (38.4%)	1543 (55.3%)	7893 (49.9%)	535 (52.4%)	<0.001
Use of clopidogrel	4729 (13.7%)	680 (24.4%)	3314 (21.0%)	216 (21.2%)	<0.001
Use of ticlopidine	560 (1.6%)	64 (2.3%)	336 (2.1%)	22 (2.2%)	<0.001
MI or CHF	14021 (40.6%)	1626 (58.3%)	8366 (52.9%)	501 (49.1%)	<0.001

Data are expressed as mean ± standard deviation for continuous variables and as numbers (percentages) for categorical variables. *p*-values are tested using a one-way analysis of variance, followed by Tukey post hoc test for continuous variables and Pearson’s χ^2^ test for categorical variables. Abbreviations: CCI, Charlson Comorbidity Index; CHF, congestive heart failure; ESRD, end-stage renal disease; MI, myocardial infarction; RASB, renin-angiotensin system blockers. * *p* < 0.05 vs. Group 1, ^#^
*p* <0.05 vs. Group 2, ^+^
*p* <0.05 vs. Group 3.

**Table 2 biomedicines-11-02838-t002:** Cox regression analyses assessing patient survival.

	Univariate		Multivariate	
	HR (95% CI)	*p*	HR (95% CI)	*p*
Group				
Ref: Group 1				
Group 2	1.10 (1.04–1.17)	0.002	1.07 (0.99–1.15)	0.108
Group 3	1.05 (1.02–1.09)	<0.001	1.01 (0.97–1.05)	0.753
Group 4	1.01 (0.92–1.12)	0.795	0.98 (0.87–1.11)	0.785
Ref: Group 2				
Group 3	0.96 (0.90–1.02)	0.171	0.94 (0.87–1.02)	0.158
Group 4	0.92 (0.82–1.03)	0.162	0.92 (0.80–1.06)	0.262
Ref: Group 3				
Group 4	0.96 (0.87–1.07)	0.467	0.98 (0.86–1.10)	0.714
Age (increase per 1 year)	1.06 (1.06–1.06)	<0.001	1.06 (1.06–1.06)	<0.001
Sex (ref: male)	0.86 (0.84–0.89)	<0.001	0.74 (0.72–0.77)	<0.001
Underlying cause of ESRD (ref: DM)	0.81 (0.80–0.82)	<0.001	0.90 (0.88–0.91)	<0.001
Vascular access (ref: AVF)	1.51 (1.46–1.56)	<0.001	1.18 (1.13–1.23)	<0.001
HD vintage (increase per 1 month)	0.99 (0.99–1.01)	0.097	1.00 (1.00–1.01)	<0.001
CCI score (increase per 1 score)	1.14 (1.13–1.14)	<0.001	1.06 (1.06–1.07)	<0.001
UFV (increase per 1 kg/session)	0.91 (0.90–0.93)	<0.001	1.07 (1.05–1.09)	<0.001
Kt/V_urea_ (increase per 1 unit)	0.91 (0.86–0.96)	<0.001	0.81 (0.76–0.88)	<0.001
Hb (increase per 1 g/dL)	0.86 (0.85–0.88)	<0.001	0.90 (0.88–0.92)	<0.001
Salb (increase per 1 g/dL)	0.37 (0.35–0.38)	<0.001	0.62 (0.59–0.65)	<0.001
SCr (increase per 1 mg/dL)	0.87 (0.86–0.87)	<0.001	0.93 (0.93–0.94)	<0.001
Sph (increase per 1 mg/dL)	0.85 (0.84–0.86)	<0.001	1.04 (1.03–1.06)	<0.001
SCa (increase per 1 mg/dL)	0.94 (0.92–0.95)	<0.001	1.06 (1.04–1.09)	<0.001
SBP (increase per 1 mmHg)	1.01 (1.01–1.01)	<0.001	1.01 (1.00–1.01)	<0.001
DBP (increase per 1 mmHg)	0.98 (0.98–0.99)	<0.001	1.00 (1.00–1.01)	0.010
Use of RASBs	1.15 (1.12–1.18)	<0.001	1.01 (0.97–1.04)	0.676
Use of statin	1.10 (1.07–1.13)	<0.001	0.93 (0.90–0.97)	<0.001
Use of ticlopidine	1.01 (0.91–1.11)	0.892	1.06 (0.95–1.18)	0.286
Use of clopidogrel	1.54 (1.49–1.59)	<0.001	1.15 (1.11–1.20)	<0.001
Use of aspirin	1.17 (1.14–1.20)	<0.001	0.96 (0.93–0.99)	0.011
MI or CHF	1.50 (1.46–1.54)	<0.001	1.05 (1.02–1.09)	0.005

Multivariate analysis was adjusted for underlying cause of ESRD, age, sex, vascular access, HD vintage, CCI score, UFV, Kt/V_urea_, Hb, Salb, SCr, Sph, SCa, SBP, DBP, use of RASBs, statin, ticlopidine, clopidogrel, and aspirin, MI or CHF. It was performed using enter mode. Abbreviations: AVF, arteriovenous fistula; CCI, Charlson Comorbidity Index; CHF, congestive heart failure; CI, confidence interval; DBP, diastolic blood pressure; DM, diabetes mellitus; ESRD, end–stage renal disease; Hb, hemoglobin; HD, hemodialysis; HR, hazard ratio; MI, myocardial infarction; RASB, renin-angiotensin system blocker; Salb, serum albumin; SBP, systolic blood pressure; SCa, serum calcium; SCr, serum creatinine; Sph, serum phosphorus; UFV, ultrafiltration volume. Group 1, patients without a prescription of β-blockers; Group 2, patients with a prescription of dialyzable and cardioselective β-blockers; Group 3, patients with a prescription of non-dialyzable and non-cardioselective β-blockers; Group 4, patients with prescription of non-dialyzable and cardioselective β-blockers.

**Table 3 biomedicines-11-02838-t003:** Cox regression analyses for patient survival based on β-blocker characteristics.

	Univariate		Multivariate	
	HR (95% CI)	*p*	HR (95% CI)	*p*
Cardioselectivity				
Ref: Group 1				
Group 3	1.05 (1.02–1.09)	<0.001	1.01 (0.97–1.05)	0.748
Group 2 or 4	1.08 (1.02–1.14)	<0.001	1.04 (0.97–1.11)	0.235
Ref: Group 3				
Group 2 or 4	1.02 (0.97–1.08)	0.432	1.04 (0.97–1.11)	0.332
Dialyzability				
Ref: Group 1				
Group 2	1.10 (1.04–1.17)	0.002	1.07 (0.99–1.15)	0.108
Group 3 or 4	1.05 (1.02–1.08)	0.001	1.01 (0.97–1.04)	0.802
Ref: Groups 2				
Group 3 or 4	0.95 (0.90–1.02)	0.149	0.94 (0.87–1.02)	0.147

Multivariate analysis was adjusted for underlying cause of end-stage renal disease, age, sex, vascular access, hemodialysis vintage, Charlson Comorbidity Index score, ultrafiltration volume, Kt/V_urea_, hemoglobin, serum albumin, serum creatinine, serum phosphorus, serum calcium, systolic blood pressure, diastolic blood pressure, use of renin-angiotensin system blockers, statin, ticlopidine, clopidogrel, and aspirin, myocardial infarction or congestive heart failure. It was performed using enter mode. Abbreviations: CI, confidence interval; HR, hazard ratio. Group 1, patients without a prescription of β-blockers; Group 2, patients with a prescription of dialyzable and cardioselective β-blockers; Group 3, patients with a prescription of non-dialyzable and non-cardioselective β-blockers; Group 4, patients with prescription of non-dialyzable and cardioselective β-blockers.

## Data Availability

The raw data were generated at the Health Insurance Review and Assessment Service. The database can be requested from the Health Insurance Review and Assessment Service by sending a study proposal including the purpose of the study, study design, and duration of analysis through an e-mail (turtle52@hira.or.kr) or at the website (https://www.hira.or.kr). The authors cannot distribute the data without permission.

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
