# Peer review of "Influence of Different Types of β-Blockers on Mortality in Patients on Hemodialysis"

_biomedicines, 2023, doi:10.3390/biomedicines11102838_

Round 1
Reviewer 1 Report
To:
Editorial Board
Biomedicines
Title: “Influence of different types of β-blockers on mortality in patients on hemodialysis”
Dear Editor,
I read this paper and I think that:
- the retrospective nature of this study is a limitation. Please discuss such a point in a dedicated limitation section.
- Please include a flow chart of the study.
- One of the main limitation when dealing with data taken by Health Insurance dataset is the lacking of fundamental data. In the supplementary file authors indicate the presence of congestive heart failure. Indeed:
a. Patients on HD may have higher incidence of amyloidosis which might mainly promote HFpEF/HFmrHF. Distinguishing type of HF is fundamental for the prognosis of patients beyond any betablockers treatment.
b. Medications had been included in the supplementary table. Indeed, dosages, relationship with dialysis, variations in treatments could not be assessed. This impact on final results and should be considered as a limitation
c. “If one or more prescription was used for a year before the evaluation of HD quality assessment program, it was considered as “use of the medication”: what about those who underwent less than one-year treatment? This is not clear.
- Authors should specifically explain primary outcome and/or possible further outcomes. They consider all-cause death: what about further “hard outcomes” which are mainly related to the use of betablockers?
- Most of comorbidities other than CV-related ones might impact on outcomes but no mention is in the multivariate regression analysis. Please discuss such a point.
Author Response
I read this paper and I think that:
- the retrospective nature of this study is a limitation. Please discuss such a point in a dedicated limitation section.
Answer: Thank you for your comments. We have included comments about the limitations of retrospective studies in the Discussion section. The added comments are as follows: This study being retrospective has the potential for selection bias, concerns about data quality, and difficulties in establishing causal relationships. Therefore, it is important to exercise caution when generalizing findings or interpreting the results.
We have included these comments in the Discussion section.
- Please include a flow chart of the study.
Answer: Thank you for your comments. We have included Figure 1 to present the flow chart of our study. Below is Figure 1.
Figure 1. Study flow chart. Abbreviations: CS, cardioselective; DB, dialyzable; HD, hemodialysis; NCS, non-cardioselective; NDB, non-dialyzable.
- One of the main limitation when dealing with data taken by Health Insurance dataset is the lacking of fundamental data. In the supplementary file authors indicate the presence of congestive heart failure. Indeed:
- Patients on HD may have higher incidence of amyloidosis which might mainly promote HFpEF/HFmrHF. Distinguishing type of HF is fundamental for the prognosis of patients beyond any betablockers treatment.
Answer: Thank you for your comments. An important limitation of our study is that we did not consider the cause and severity of heart failure. We lacked data on these aspects, which is inherent to our study design that diagnoses the disease using a different code-based approach, rather than relying on data like chart review, laboratory, and imaging data. In this study, our approach to heart failure was limited in terms of detailed cause assessment and the ability to assess the degree of functional impairment. Heart failure has various potential causes, such as ischemia, blood pressure, arrhythmia, or amyloidosis, and can be divided into three groups: heart failure with reduced ejection fraction, heart failure with moderately reduced ejection fraction, and heart failure with preserved ejection rejection. These factors would influence the prognosis and impact on clinical outcomes depending on the type of β-blockers. We consider this a significant limitation of our study. We believe that addressing these limitations will be essential through comprehensive subgroup analyses and other means in future research endeavors.
These comments have been added in the Discussion section.
- Medications had been included in the supplementary table. Indeed, dosages, relationship with dialysis, variations in treatments could not be assessed. This impact on final results and should be considered as a limitation
Answer: Thank you for your valuable comments. Our study lacks data on specific medication dosages, timing of administration in relation to dialysis sessions, and self-adjustment of medication based on changes in blood pressure. There might be differences between the prescribed amount of medication and the actual dosage taken, particularly among patients on dialysis who may take medications after dialysis due to intradialytic hypotension or immediately before dialysis in cases of intradialytic hypertension. Additionally, patients may voluntarily discontinue medications if their blood pressure is low or take additional doses when it is high. It is important to consider these factors as confounding variables that could potentially affect our results. However, the limitation of our study is the lack of detailed information on how medication was administered.
We have included these comments in the Discussion section.
- “If one or more prescription was used for a year before the evaluation of HD quality assessment program, it was considered as “use of the medication”: what about those who underwent less than one-year treatment? This is not clear.
Answer: Thank you for your comments. In our study, we determined if a patient had used aspirin, renin-angiotensin system blockers, clopidogrel, ticlopidine, or statin by checking if they had a filled prescription for medication that was utilized within the year before the evaluation of the HD quality assessment program. For patients who underwent HD for less than a year, we applied a similar criterion but adjusted it according to the duration of their treatment. Specifically, we checked if they had a filled prescription for the medication that was utilized during their treatment period, which could be shorter than a year.
We have updated the relevant comments.
- Authors should specifically explain primary outcome and/or possible further outcomes. They consider all-cause death: what about further “hard outcomes” which are mainly related to the use of betablockers?
Answer: Thank you for your comments.
We have included data for cardiac-cerebrovascular events (CVE) in our study. We evaluated all-cause mortality as the primary outcome and CVE as the secondary outcome. In our study, CVE was defined as interventions associated with revascularization, which were evaluated using medical treatment, procedure, or operation codes, as shown in Table S3.
Table S3. Codes associated with cardiac and cerebrovascular outcomes.
|
Procedure or operation codes |
|
|
Percutaneous coronary intervention |
M6551, M6552, M6561~M6564, M6571, M6572, M6601, M6602 |
|
Coronary artery bypass grafting |
O1641, O1642, O1647, OA641, OA642, OA647 |
|
Medical treatment codes |
|
|
Protein C |
635801BIJ |
|
Tissue type plasminogen activator |
223501BIJ, 223502BIJ |
|
Tenecteplase |
450302BIJ, 450301BIJ |
|
Tirofiban |
240201BIJ, 240230BIJ |
|
Urokinase |
246401BIJ, 246405BIJ, 246407BIJ, 246404BIJ, 246406BIJ |
Mortality was evaluated for all patients. For the analyses of CVE, we excluded patients who had interventions for CVE within a year before and during HD quality assessment. The analysis of CVE included 31,080 patients in Group 1, 2,353 patients in Group 2, 14,129 patients in Group 3, and 887 patients in Group 4. The Kaplan–Meier curves for CVE according to the groups are presented in Figure S2.
Figure S2. Kaplan–Meier curves depicting CVE-free survival based on different groups. The P-values for pairwise comparison or trend with log-rank tests are added to the upper right corner of the graph. Abbreviations: CVE, cardiac and cerebrovascular events; Group 1, patients without prescription of β-blockers; Group 2, patients with prescription of dialyzable and cardioselective β-blockers; Group 3, patients with prescription of non-dialyzable and non-cardioselective β-blockers; Group 4, patients with prescription of non-dialyzable and cardioselective β-blockers
The 5-year survival rates in Groups 1, 2, 3, and 4 were 86.3%, 83.6%, 85.1%, and 85.1%, respectively. In univariate and multivariate analyses, Group 2 had poorer CVE-free survival rates than Group 1 or 3 (Table S5). Group 2 showed a trend of poorer CVE-free survival rates than did Group 4, althouth it was not statistically significant.
Table S5. Cox regression analyses for CVE
|
|
Univariate |
Multivariate |
||
|
HR (95% CI) |
P |
HR (95% CI) |
P |
|
|
Group |
|
|
|
|
|
Ref: Group 1 |
|
|
|
|
|
Group 2 |
1.23 (1.10–1.37) |
<0.001 |
1.15 (1.00–1.31) |
0.048 |
|
Group 3 |
1.09 (1.03–1.15) |
0.002 |
0.99 (0.92–1.05) |
0.682 |
|
Group 4 |
1.08 (0.91–1.30) |
0.381 |
1.06 (0.87–1.31) |
0.550 |
|
Ref: Group 2 |
|
|
|
|
|
Group 3 |
0.89 (0.79–0.99) |
0.035 |
0.86 (0.75–0.99) |
0.034 |
|
Group 4 |
0.88 (0.72–1.08) |
0.232 |
0.93 (0.73–1.18) |
0.548 |
|
Ref: Group 3 |
|
|
|
|
|
Group 4 |
1.00 (0.83–1.19) |
0.966 |
1.08 (0.88–1.33) |
0.469 |
Multivariate analysis was adjusted for several variables including age, sex, underlying cause of end-stage renal disease, vascular access, hemodialysis vintage, Charlson Comorbidity Index score, ultrafiltration volume, Kt/Vurea, hemoglobin, serum albumin, serum creatinine, serum phosphorus, serum calcium, systolic blood pressure, diastolic blood pressure, use of renin-angiotensin system blockers, statin, ticlopidine, clopidogrel, and aspirin, myocardial infarction or congestive heart failure. It was performed using enter mode. Group 1, patients without prescription of β-blockers; Group 2, patients with prescription of dialyzable and cardioselective β-blockers; Group 3, patients with prescription of non-dialyzable and non-cardioselective β-blockers; Group 4, patients with prescription of non-dialyzable and cardioselective β-blockers.
In analyses involving patients who did not receive treatment for recent cardiac or cerebrovascular events, Group 2 had a lower CVE-free survival rate than Group 1 or 3. However, there was no association with all-cause mortality. Additionally, there was a worsening trend in CVE-free survival for Group 2 when compared to Group 4. The lack of statistical significance between Group 2 and Group 4 may be attributed to the small sample size of Group 4. The findings of this study are consistent with Weir’s research, which demonstrated that non-dialyzable β-blockers yielded better results than dialyzable β-blockers. The results imply that, for patients who have not undergone recent cardiac or cerebrovascular interventions, the positive effects of dialyzable β-blockers on intradialytic hypotension are diminished, and the risk of adverse effects, such as hypertension caused by the removal of the drug, may be amplified. Patients who have not undergone recent cardiac or cerebrovascular interventions may be prescribed β-blockers for hypertension. Additionally, non-cardioselective β-blockers have a greater effect on lowering blood pressure compared to cardioselective β-blockers. Removal of the drug during HD could potentially worsen blood pressure during HD sessions and increase the risk of CVE compared to patients taking non-dialylzable β-blockers. However, it is important to interpret these findings cautiously due to the limitations of our study and the lack of significant association with all-cause mortality.
We have included these comments in the Methods, Results, and Discussion sections.
- Most of comorbidities other than CV-related ones might impact on outcomes but no mention is in the multivariate regression analysis. Please discuss such a point.
Answer: Thank you for your comments. The prognosis of patients on HD can be significantly influenced by various comorbidities, so adjusting for these factors is essential in multivariate analyses. Instead of adjusting individually for each comorbidity, we chose a simplified approach in our study. Specifically, we used a merged score from the Charlson Comorbidity Index (CCI) as a confounding factor in multivariate analyses. We also conducted separate analyses for DM and heart diseases (MI or CHF). The CCI in our study included 17 comorbidities: MI, CHF, peripheral vascular disease, cerebrovascular disease, dementia, chronic pulmonary disease, rheumatologic disease, peptic ulcer disease, mild liver disease, DM without chronic complications, hemiplegia, renal disease, DM with chronic complications, any malignancy, moderate to severe liver disease, metastatic tumor, and acquired immune deficiency syndrome. Among these comorbidities, only heart disease (MI or CHF) was used as an independent confounding factor or included in subgroup analysis. DM was included in subgroup analysis. However, for other comorbidities, we only used them as confounding factors in multivariate analyses, using the CCI scores derived from them. While each comorbidity included in the CCI might have an individual impact on the prognosis of patients on HD, our study focused more on differences in overall mortality rates based on types of β-blockers. Therefore, we opted for a simpler model. Analyzing the individual risks associated with each comorbidity is an interesting research topic, but it was not the main objective of our study.
We have included these comments in the Discussion section.

Reviewer 2 Report
The present study is based on numerous data collected in HD patients, and is about the significance of the type of betablocker (BB) used in influencing outcomes. The design and methodology of the study are well done, but unfortunately, we don`t know anything about the three main indications of BB therapy in the patients studied: CAD, HFrEF and atrial fibrillation. Without these data is questionable the meaning of the whole study, because finding the right BB for the right patient, maybe, is the most important goal of such a study. Speculating hemodynamic data related to "black box" HD patients is also questionable. I think, the study in this form is well done, but not informative for the practician. I would like to have a more nuanced explanation on the differential use of BBs in HD patients, in daily practice, based on the authors own experience and the data from the literature.
minor corrections are needed
Author Response
The present study is based on numerous data collected in HD patients, and is about the significance of the type of betablocker (BB) used in influencing outcomes. The design and methodology of the study are well done, but unfortunately, we don`t know anything about the three main indications of BB therapy in the patients studied: CAD, HFrEF and atrial fibrillation. Without these data is questionable the meaning of the whole study, because finding the right BB for the right patient, maybe, is the most important goal of such a study. Speculating hemodynamic data related to "black box" HD patients is also questionable. I think, the study in this form is well done, but not informative for the practician. I would like to have a more nuanced explanation on the differential use of BBs in HD patients, in daily practice, based on the authors own experience and the data from the literature.
Answer: Thank you for your comments. We have included the results of patients with myocardial infarction, congestive heart failure, or atrial fibrillation in Table S4. Additionally, we have provided some practical information regarding the use of β-blockers in patients on HD. The majority of our findings indicate that there is no association between the use or type of β-blockers and mortality.
Table S4. Cox regression analyses for mortality in patients with MI, CHF, or atrial fibrillation.
|
|
Univariate |
Multivariate |
||
|
HR (95% CI) |
P |
HR (95% CI) |
P |
|
|
MI |
|
|
|
|
|
Ref: Group 1 |
|
|
|
|
|
Group 2 |
0.93 (0.79–1.09) |
0.348 |
0.94 (0.76–1.17) |
0.590 |
|
Group 3 |
0.94 (0.86–1.03) |
0.184 |
0.98 (0.88–1.11) |
0.789 |
|
Group 4 |
0.72 (0.72–1.23) |
0.643 |
0.88 (0.62–1.26) |
0.487 |
|
Ref: Group 2 |
|
|
|
|
|
Group 3 |
1.02 (0.86–1.20) |
0.839 |
1.04 (0.84–1.30) |
0.700 |
|
Group 4 |
1.01 (0.75–1.37) |
0.935 |
0.93 (0.62–1.40) |
0.744 |
|
Ref: Group 3 |
|
|
|
|
|
Group 4 |
1.00 (0.76–1.31) |
0.973 |
0.89 (0.62–1.28) |
0.544 |
|
CHF |
|
|
|
|
|
Ref: Group 1 |
|
|
|
|
|
Group 2 |
1.02 (0.94–1.10) |
0.631 |
1.08 (0.98–1.20) |
0.125 |
|
Group 3 |
0.93 (0.90–0.98) |
0.002 |
0.97 (0.91–1.02) |
0.198 |
|
Group 4 |
0.97 (0.85–1.12) |
0.682 |
0.93 (0.78–1.11) |
0.407 |
|
Ref: Group 2 |
|
|
|
|
|
Group 3 |
0.92 (0.84–0.99) |
0.040 |
0.89 (0.80–0.99) |
0.031 |
|
Group 4 |
0.95 (0.82–1.11) |
0.540 |
0.86 (0.70–1.04) |
0.124 |
|
Ref: Group 3 |
|
|
|
|
|
Group 4 |
1.04 (0.90–1.20) |
0.588 |
0.96 (0.80–1.15) |
0.664 |
|
Atrial fibrillation |
|
|
|
|
|
Ref: Group 1 |
|
|
|
|
|
Group 2 |
1.14 (1.00–1.31) |
0.058 |
1.09 (0.91–1.30) |
0.336 |
|
Group 3 |
0.90 (0.82–0.98) |
0.019 |
0.94 (0.84–1.06) |
0.323 |
|
Group 4 |
0.89 (0.68–1.15) |
0.372 |
0.85 (0.60–1.21) |
0.362 |
|
Ref: Group 2 |
|
|
|
|
|
Group 3 |
0.79 (0.68–0.91) |
0.002 |
0.86 (0.72–1.04) |
0.131 |
|
Group 4 |
0.78 (0.58–1.04) |
0.086 |
0.78 (0.53–1.14) |
0.199 |
|
Ref: Group 3 |
|
|
|
|
|
Group 4 |
0.99 (0.76–1.30) |
0.931 |
0.90 (0.63–1.29) |
0.567 |
Multivariate analysis was adjusted for various factors, including age, sex, underlying cause of end-stage renal disease, vascular access, hemodialysis vintage, Charlson Comorbidity Index score, ultrafiltration volume, Kt/Vurea, hemoglobin, serum albumin, serum creatinine, serum phosphorus, serum calcium, systolic blood pressure, diastolic blood pressure, use of renin-angiotensin system blockers, statin, ticlopidine, clopidogrel, and aspirin, MI or CHF. It was performed using enter mode. MI or CHF as covariate was excluded in multivariate analyses using subgroups with MI or CHF. Abbreviations: CHF, congestive heart failure; CI, confidence interval; HR, hazard ratio; MI, myocardial infarction. Group 1, patients without prescription of β–blockers; Group 2, patients with prescription of dialyzable and cardioselective β–blockers; Group 3, patients with prescription of non-dialyzable and non-cardioselective β–blockers; Group 4, patients with prescription of non-dialyzable and cardioselective β–blockers.
We conducted subgroup analyses on patients with coexisting conditions, such as myocardial infarction (MI), congestive heart failure (CHF), and atrial fibrillation, to investigate the effects of β-blockers and any potential differences among β-blockers. However, our study did not show any survival advantage associated with the use of β-blockers or significant differences between the various types. Considering previous studies and the specific limitations and characteristics of our research, we initially anticipated that the use of β-blocker in patients on HD with conditions such as MI, CHF, and atrial fibrillation would likely have a positive impact on patient prognosis. The lack of findings in our study may be due to limitations and patient characteristics. We acknowledge that using of ICD-10 codes for diagnosis may introduce inaccuracies, as accurate diagnosis usually requires essential tests like coronary angiography, electrocardiography, and echocardiography. Additionally, the high prevalence of comorbidities and the relatively advanced age of patients on HD likely had a significant impact on the potential side effects of β-blockers. This impact could have reduced the overall benefit of using β-blocker. Therefore, when considering the use of β-blocker in patients on HD with appropriate indications, it is essential to carefully assess and potentially modify their use based on associated complications.
In light of our study's findings on β-blocker use, type differences, analyses across various subgroups, and existing literature, we conclude that patients on HD show distinct variations. Currently, there is insufficient evidence to definitively prove the superiority of any specific β-blocker type. Decisions regarding β-blocker use should be based on pharmacodynamics and personalized approaches. While the advantages of β-blockers, particularly cardioselective ones, are considered in conditions such as MI, CHF, and atrial fibrillation, it is rational to consider factors like blood pressure fluctuations, heart rate, and the timing of these changes when determining the continued use or potential switch of β-blocker types. Furthermore, if the main objective is blood pressure control, non-cardioselective β-blockers might be given priority. In patients on HD, when the benefits may not be significantly greater than those in patients not on dialysis due to various underlying conditions, it may not be necessary to hesitate when discontinuing β-blockers if they cause side effects that could potentially worsen the prognosis. Nevertheless, we did observe differences in mortality rates related to the type of β-blockers in certain subgroups, particularly the elderly, those with diabetes mellitus (DM), and heart diseases. Among these groups, non-dialyzable non-cardioselective β-blockers showed a more favorable prognosis compared to dialyzable β-blockers, possibly due to fewer hemodynamic changes.
These comments have been included in the Results and Discussion sections.

Reviewer 3 Report
In Line 88: I suggest to replace "We divided them..." into "We divided included patients..."
Author Response
In Line 88: I suggest to replace "We divided them..." into "We divided included patients..."
Answer: Thank you for these comments. We have accordingly revised the mentioned sentence.

Round 2
Reviewer 1 Report
authors well addressed my previous comments. The paper improved.
Reviewer 2 Report
The manuscript was properly updated and improved.
none